# Estimation of Temperature and Associated Uncertainty from Fiber-Optic Raman-Spectrum Distributed Temperature Sensing

**DOI:** 10.3390/s20082235

**Published:** 2020-04-15

**Authors:** Bas des Tombe, Bart Schilperoort, Mark Bakker

**Affiliations:** Water Resources Section, Faculty of Civil Engineering and Geosciences, Delft University of Technology, Stevinweg 1, 2628CN Delft, The Netherlands; B.Schilperoort@tudelft.nl (B.S.); mark.bakker@tudelft.nl (M.B.)

**Keywords:** distributed temperature sensing, DTS, fiber optic, Raman, Stokes, temperature, calibration, uncertainty, confidence intervals

## Abstract

Distributed temperature sensing (DTS) systems can be used to estimate the temperature along optic fibers of several kilometers at a sub-meter interval. DTS systems function by shooting laser pulses through a fiber and measuring its backscatter intensity at two distinct wavelengths in the Raman spectrum. The scattering-loss coefficients for these wavelengths are temperature-dependent, so that the temperature along the fiber can be estimated using calibration to fiber sections with a known temperature. A new calibration approach is developed that allows for an estimate of the uncertainty of the estimated temperature, which varies along the fiber and with time. The uncertainty is a result of the noise from the detectors and the uncertainty in the calibrated parameters that relate the backscatter intensity to temperature. Estimation of the confidence interval of the temperature requires an estimate of the distribution of the noise from the detectors and an estimate of the multi-variate distribution of the parameters. Both distributions are propagated with Monte Carlo sampling to approximate the probability density function of the estimated temperature, which is different at each point along the fiber and varies over time. Various summarizing statistics are computed from the approximate probability density function, such as the confidence intervals and the standard uncertainty (the estimated standard deviation) of the estimated temperature. An example is presented to demonstrate the approach and to assess the reasonableness of the estimated confidence intervals. The approach is implemented in the open-source Python package “dtscalibration”.

## 1. Introduction

Temperatures can be measured with distributed temperature sensing (DTS) along optical fibers that may extend to several kilometers with a sub-meter resolution [1,2]. The application of DTS is of great value to characterize thermal dynamics at a scale that corresponds to, for example, many geophysical processes. Heat tracer tests with DTS have been used to estimate wind speed [3,4], evaporation [5,6], soil moisture [7], groundwater-surface water interaction [1,8], and groundwater flow [9,10]. The specification sheets of DTS measurement systems list the uncertainty in the temperature estimated under ideal conditions. In practice, the uncertainty in the temperature estimate varies along the fiber, is different for each setup, and is temperature-dependent. Therefore, the uncertainty in the temperature estimate should be estimated separately for every DTS measurement.

DTS systems that estimate temperature from Raman-backscatter measurements shoot a laser pulse through a fiber-optic cable, which is scattered back to the DTS system by inhomogeneities in the fiber. Most of the backscattered laser has the same wavelength as the emitted laser (Rayleigh scattering), but a small fraction has different wavelengths (Raman scattering). The detectors in DTS systems measure the intensity of the backscatter at two distinct wavelengths: Stokes (-Raman) and anti-Stokes (-Raman) scatter. The temperature at the point of reflection is estimated from these two types of scatter. Stokes scatter has a longer wavelength than the laser and its intensity does not vary much with temperature, while anti-Stokes scatter has a shorter wavelength than the laser and its intensity varies significantly with temperature. The location of the measurement along the fiber is estimated from the time between sending the laser pulse and receiving the scatter. Temperature along the fiber is estimated from the measured intensities of the Stokes and anti-Stokes scatter by calibrating to reference sections with a known temperature. In practice, these fiber sections are submerged in water baths of which the temperature is continuously measured with a separate temperature sensor. The water can be mixed with small pumps in an attempt to equalize the temperature of the water in the baths. Sequential temperature measurements require continuous calibration due to varying gains and losses in the DTS system. Detailed calibration procedures are available in the literature [11,12,13,14]. The uncertainty in the temperature estimates is strongly affected by the calibration procedure [2,15].

Attenuation of light propagating through a fiber depends on its wavelength and therefore is different for Stokes and anti-Stokes scatter. The key in DTS calibration is to differentiate the attenuation from the temperature effects at the point of reflection. There are two types of setups to estimate temperature from Stokes and anti-Stokes scatter: single-ended and double-ended. In single-ended setups, only one end of the fiber is connected to the DTS system, and the difference in attenuation between the Stokes and the anti-Stokes scatter is approximated as constant over the fiber. However, this difference in attenuation is also affected by sharp bends, connectors, and bad splices, which may result in a shift in the temperature if not accounted for. In double-ended setups, measurements are carried out from both ends of the fiber to estimate the difference in attenuation between the Stokes and anti-Stokes scatter at each point along the fiber. In such setups, measurements from both ends of the fiber must be combined, including their uncertainty.

In this paper, a new calibration procedure is presented for the temperature estimates for single-ended and double-ended setups, including estimates of the uncertainty in the form of confidence intervals. First, a brief review is given of how temperature can be estimated from Stokes and anti-Stokes intensity measurements. Next, the calibration steps are outlined for single-ended and double-ended setups. The spatial and temporal variability in the uncertainty of the temperature are demonstrated by an example of the double-ended calibration procedure. The new calibration procedure is implemented in the open-source Python package “dtscalibration”. This article concludes with a conclusion and a discussion with practical tips to reduce the uncertainty in the temperature.

## 2. Estimation of Temperature from Stokes and Anti-Stokes Scatter

The equations that relate the Stokes and anti-Stokes intensity measurements to temperature along a fiber are reviewed here briefly; a comprehensive resource that covers DTS theory is Hartog [16]. The path of a laser pulse emitted by a DTS system through a fiber is shown in Figure 1. The laser module emits a short laser pulse that is approximated here as an impulse with energy $E_{0}$. The pulse travels with the speed of light, *v*, through the fiber, during which it attenuates with $\exp\left(-\int_{0}^{x}\alpha_{l}\;dx\right)$, where $\alpha_{l}$ is the attenuation at the wavelength of the laser pulse. A small fraction, S(x)R(x)Δx, of the energy of the laser pulse is scattered back over section Δx, where *S* is the scattering-loss coefficient for the scattering at the wavelength of the laser pulse, and *R* is the capture fraction, which depends on the fiber composition (e.g., diameter and numerical aperture [17]). The backscatter pulse is 2Δt wide and is attenuated again on its way back to the detector. The intensity measured by the detector, *P* from scattering at *x* (between times 2x/v−Δt and 2x/v+Δt) is given by [16] (Equation (3.5)):(1)$$\begin{array}{cc} P\left(x,t\right) & =\frac{1}{2}E_{0}\eta vSR\exp\left(-\int_{0}^{x}2\alpha_{l}\;dx\right) \end{array}$$
where η corrects for the sensitivity of the detector and the attenuation between the detector and where the fiber end is connected to the DTS system [18].

In practice, the backscatter is not returned as a pulse, but is dispersed, resulting in spatially correlated measurements. The spreading of the pulse is caused by the finite width of the emitted laser pulse, the measurement response of the detector, and intermodal dispersion of the pulse while propagating through the fiber. The spreading of the pulse may be experimentally estimated following Simon et al. [19]. Most of the scattered energy has the same wavelength as the emitted laser pulse (Rayleigh scattering), but a small part has different wavelengths. The scattering-loss coefficient $S_{+}$ for the Stokes wavelength and $S_{-}$ for the anti-Stokes wavelength are given by [16,20]: (2)$$\begin{array}{c} S_{+}\left(x,t\right)=\frac{K_{+}}{\lambda_{+}^{4}}\frac{\exp\left[\gamma /T(x,t)\right]}{\exp\left[\gamma /T(x,t)\right]-1} \end{array}$$
(3)$$\begin{array}{c} S_{-}\left(x,t\right)=\frac{K_{-}}{\lambda_{-}^{4}}\frac{1}{\exp\left[\gamma /T(x,t)\right]-1} \end{array}$$
where $K_{+}$ and $K_{-}$ correct for the fraction of the molecules in the fiber that scatter at the Stokes and anti-Stokes wavelengths [21], λ is the wavelength, *T* is the temperature along the fiber in Kelvin, and γ is the sensitivity of the Stokes and anti-Stokes scattering to temperature in Kelvin and depends on the fiber material.

The Stokes and anti-Stokes power ($P_{+}$ and $P_{-}$, respectively) measured at the detector(s) in a DTS system are given by: (4)$$\begin{array}{cc} P_{+}\left(x,t\right) & =\frac{1}{2}E_{0}\eta_{+}vS_{+}R\exp\left[-\int_{0}^{x}\left(\alpha_{l}+\alpha_{+}\right)\;dx\right] \end{array}$$
(5)$$\begin{array}{cc} P_{-}\left(x,t\right) & =\frac{1}{2}E_{0}\eta_{-}vS_{-}R\exp\left[-\int_{0}^{x}\left(\alpha_{l}+\alpha_{-}\right)\;dx\right] \end{array}$$
where $\alpha_{+}$ and $\alpha_{-}$ are the attenuation as a function of *x* at the Stokes and anti-Stokes wavelengths. Most DTS systems use the ratio of Equation (Equation 4) over Equation (Equation 5) [20]:(6)$$\begin{array}{cc} \frac{P_{+}\left(x,t\right)}{P_{-}\left(x,t\right)} & =\frac{\eta_{+}\left(t\right)K_{+}/\lambda_{+}^{4}}{\eta_{-}\left(t\right)K_{-}/\lambda_{-}^{4}}\exp\left(-\int_{0}^{x}\Delta \alpha \left(x^{\prime }\right)dx^{\prime }\right)\exp\left[\gamma /T(x,t)\right] \end{array}$$
where Equations (Equation 2) and (Equation 3) are substituted for $S_{+}$ and $S_{-}$, respectively, and Δα
$\left(m^{-1}\right)$ is the differential attenuation, $\Delta \alpha \left(x\right)=\alpha_{+}-\alpha_{-}$. Taking the logarithm of Equation (Equation 6) gives:(7)$$\begin{array}{cc} I(x,t) & =-C\left(t\right)-\int_{0}^{x}\Delta \alpha \left(x^{\prime }\right)dx^{\prime }+\frac{\gamma }{T(x,t)} \end{array}$$
where
(8)$$\begin{array}{cc} I(x,t) & =\ln\left(\frac{P_{+}\left(x,t\right)}{P_{-}\left(x,t\right)}\right) \end{array}$$
(9)$$\begin{array}{cc} C(t) & =\ln\left(\frac{\eta_{-}\left(t\right)K_{-}/\lambda_{-}^{4}}{\eta_{+}\left(t\right)K_{+}/\lambda_{+}^{4}}\right) \end{array}$$
where *C* is the lumped effect of the difference in gain at x=0 between Stokes and anti-Stokes intensity measurements and the dependence of the scattering intensity on the wavelength. An equation for *T* may be obtained from Equation (Equation 7) as:(10)$$\begin{array}{c} T\left(x,t\right)=\frac{\gamma }{I\left(x,t\right)+C\left(t\right)+\int_{0}^{x}\Delta \alpha \left(x^{\prime }\right)dx^{\prime }} \end{array}$$

The temperature along the fiber can now be estimated from the Stokes and anti-Stokes intensity measurements, *I*, if the terms γ, *C*, and $\int_{0}^{x}\Delta \alpha \left(x^{\prime }\right)dx^{\prime }$ are known. These terms are estimated by calibration to reference sections.

## 3. Integrated Differential Attenuation

The differential attenuation, Δα, is different for each fiber type, varies along a fiber, and may change at sharp fiber bends and fiber connections. The integrated differential attenuation ($\int_{0}^{x}\Delta \alpha \left(x^{\prime }\right)dx^{\prime }$) differs per setup and must be estimated experimentally to differentiate it from a shift in the temperature. It is estimated differently in single-ended setups than in double-ended setups.

### 3.1. Single-Ended Measurements

In single-ended setups, Stokes and anti-Stokes intensity is measured from a single end of the fiber. The differential attenuation is assumed constant along the fiber so that the integrated differential attenuation may be written as [11]:(11)$$\begin{array}{c} \int_{0}^{x}\Delta \alpha \left(x^{\prime }\right)dx^{\prime }\approx \Delta \alpha x \end{array}$$

The temperature can now be written from Equation (Equation 10) as:(12)$$\begin{array}{c} T\left(x,t\right)\approx \frac{\gamma }{I(x,t)+C(t)+\Delta \alpha x} \end{array}$$

The parameters γ, C(t), and Δα must be estimated from calibration to reference sections, as discussed in Section 5. The parameter *C* must be estimated for each time and is constant along the fiber. Confidence intervals for the estimated temperature are discussed in Section 7.1. When jumps in the integrated differential attenuation are expected, the measurements can be split into sections with a constant differential attenuation. Each additional section requires its own reference section to estimate the magnitude of the jump in the integrated differential attenuation.

### 3.2. Double-Ended Measurements

In double-ended setups, Stokes and anti-Stokes intensity is measured in two directions from both ends of the fiber. The forward-channel measurements are denoted with subscript F, and the backward-channel measurements are denoted with subscript B. Both measurement channels start at a different end of the fiber and have opposite directions, and therefore have different spatial coordinates. The first processing step with double-ended measurements is to align the measurements of the two measurement channels so that they have the same spatial coordinates. The spatial coordinate *x* (m) is defined here positive in the forward direction, starting at 0 where the fiber is connected to the forward channel of the DTS system; the length of the fiber is *L*. Consequently, the backward-channel measurements are flipped and shifted to align with the forward-channel measurements. Alignment of the measurements of the two channels is prone to error because it requires the exact fiber length [15]. Depending on the DTS system used, the forward channel and backward channel are measured one after another by making use of an optical switch, so that only a single detector is needed. However, it is assumed in this paper that the forward channel and backward channel are measured simultaneously, so that the temperature of both measurements is the same. This assumption holds better for short acquisition times with respect to the time scale of the temperature variation, and when there is no systematic difference in temperature between the two channels. The temperature may be computed from the forward-channel measurements (Equation (Equation 10)) with:(13)$$\begin{array}{cc} T_{F}\left(x,t\right) & =\frac{\gamma }{I_{F}\left(x,t\right)+C_{F}\left(t\right)+\int_{0}^{x}\Delta \alpha \left(x^{\prime }\right)dx^{\prime }} \end{array}$$
and from the backward-channel measurements with:(14)$$\begin{array}{cc} T_{B}\left(x,t\right) & =\frac{\gamma }{I_{B}\left(x,t\right)+C_{B}\left(t\right)+\int_{x}^{L}\Delta \alpha \left(x^{\prime }\right)dx^{\prime }} \end{array}$$
where $C_{F}\left(t\right)$ and $C_{B}\left(t\right)$ are the parameter C(t) for the forward-channel and backward-channel measurements, respectively. $C_{B}\left(t\right)$ may be different from $C_{F}\left(t\right)$ due to differences in gain, and difference in the attenuation between the detectors and the point the fiber end is connected to the DTS system ($\eta_{+}$ and $\eta_{-}$ in Equation (Equation 9)). The calibration procedure presented in van de Giesen et al. [12] approximates C(t) to be the same for the forward and backward-channel measurements, but this approximation is not made here.

Parameter A(x) is introduced to simplify the notation of the double-ended calibration procedure and represents the integrated differential attenuation between locations $x_{1}$ and *x* along the fiber. Location $x_{1}$ will be selected in Section 6.
(15)$$\begin{array}{cc} A(x) & =\int_{x_{1}}^{x}\Delta \alpha \left(x^{\prime }\right)dx^{\prime } \end{array}$$
so that Equations (Equation 13) and (Equation 14) may be written as: (16)$$\begin{array}{cc} T_{F}\left(x,t\right) & =\frac{\gamma }{I_{F}\left(x,t\right)+D_{F}\left(t\right)+A\left(x\right)} \end{array}$$
(17)$$\begin{array}{cc} T_{B}\left(x,t\right) & =\frac{\gamma }{I_{B}\left(x,t\right)+D_{B}\left(t\right)-A\left(x\right)} \end{array}$$
where
(18)$$\begin{array}{cc} D_{F}\left(t\right) & =C_{F}\left(t\right)+\int_{0}^{x_{1}}\Delta \alpha \left(x^{\prime }\right)dx^{\prime } \end{array}$$
(19)$$\begin{array}{cc} D_{B}\left(t\right) & =C_{B}\left(t\right)+\int_{x_{1}}^{L}\Delta \alpha \left(x^{\prime }\right)dx^{\prime } \end{array}$$

Parameters $D_{F}$ and $D_{B}$ must be estimated for each time and are constant along the fiber, and parameter *A* must be estimated for each location and is constant over time. The calibration procedure is discussed in Section 6. $T_{F}$ and $T_{B}$ are separate approximations of the same temperature at the same time. The estimated $T_{F}$ is more accurate near x=0 because that is where the signal is strongest. Similarly, the estimated $T_{B}$ is more accurate near x=L. A single best estimate of the temperature is obtained from the weighted average of $T_{F}$ and $T_{B}$ as discussed in Section 7.2.

## 4. Estimation of the Variance of the Noise in the Intensity Measurements

The Stokes and anti-Stokes intensities are measured with detectors, which inherently introduce noise to the measurements. Knowledge of the distribution of the measurement noise is needed for a calibration with weighted observations (Section 5 and Section 6) and to project the associated uncertainty to the temperature confidence intervals (Section 7). Two sources dominate the noise in the Stokes and anti-Stokes intensity measurements [16] (p. 125). Close to the laser, noise from the conversion of backscatter to electricity dominates the measurement noise. The detecting component, an avalanche photodiode, produces Poisson-distributed noise with a variance that increases linearly with the intensity. The Stokes and anti-Stokes intensities are commonly much larger than the standard deviation of the noise, so that the Poisson distribution can be approximated with a Normal distribution with a mean of zero and a variance that increases linearly with the intensity. At the far-end of the fiber, noise from the electrical circuit dominates the measurement noise. It produces Normal-distributed noise with a mean of zero and a variance that is independent of the intensity. However, in this paper the sum of the two noise sources is approximated with a single Normal distribution with a variance that is independent of the intensity. This approximation holds better for small setups with little attenuation and DTS systems with avalanche photodiodes with a lower gain. In Appendix A, a procedure is presented to compute the intensity-dependent variance of the noise for when this approximation does not hold.

The variance of the noise in the Stokes ($\sigma_{P_{+}}^{2}$) and anti-Stokes ($\sigma_{P_{-}}^{2}$) measurements are estimated by fitting Equations (Equation 4) and (Equation 5) through the Stokes and anti-Stokes intensity measurements of the reference sections, after which the distribution of the residuals is used as an estimate of the distribution of the noise [22]. Fiber sections that are used for calibration have an approximately spatially uniform reference temperature ($T_{r}\left(t\right)$), so that Equations (Equation 4) and (Equation 5) are each expressed in a term that varies along the reference section but is constant over time (H(x)) and a term that varies with time (G(t)) but is constant for a reference section: (20)$$\begin{array}{cc} P_{+}\left(x,t\right) & =G_{+}\left(t\right)H_{+}\left(x\right) \end{array}$$
(21)$$\begin{array}{cc} P_{-}\left(x,t\right) & =G_{-}\left(t\right)H_{-}\left(x\right) \end{array}$$
with
(22)$$\begin{array}{cc} G_{+}\left(t\right) & =E_{0}\left(t\right)\eta_{+}\left(t\right)\frac{K_{+}}{\lambda_{+}^{4}}\frac{\exp\left[\gamma /T_{r}\left(t\right)\right]}{\exp\left[\gamma /T_{r}\left(t\right)\right]-1} \end{array}$$
(23)$$\begin{array}{cc} G_{-}\left(t\right) & =E_{0}\left(t\right)\eta_{-}\left(t\right)\frac{K_{-}}{\lambda_{-}^{4}}\frac{1}{\exp\left[\gamma /T_{r}\left(t\right)\right]-1} \end{array}$$
(24)$$\begin{array}{cc} H_{+}\left(x\right) & =\frac{1}{2}vR\exp\left(-\int_{0}^{x}\left[\alpha_{l}\left(x^{\prime }\right)+\alpha_{+}\left(x^{\prime }\right)\right]dx^{\prime }\right) \end{array}$$
(25)$$\begin{array}{cc} H_{-}\left(x\right) & =\frac{1}{2}vR\exp\left(-\int_{0}^{x}\left[\alpha_{l}\left(x^{\prime }\right)+\alpha_{-}\left(x^{\prime }\right)\right]dx^{\prime }\right) \end{array}$$

Values for G(t) and H(x) are obtained by minimizing the residuals between the Stokes and anti-Stokes intensity measurements and Equations (Equation 20) and (Equation 21) with a standard routine. Here, the “minimize” routine of the Python package scipy.optimize [23] is used, and is implemented in the package “dtscalibration” (Section 8). G(t) and H(x) are, of course, highly correlated (Equations (Equation 20) and (Equation 21)), but that is not relevant here as only the product of G(t) and H(x) is of interest. The residuals between the fitted product of G(t) and H(x) and the Stokes and anti-Stokes intensity measurements are attributed to the noise from the detectors. The variance of the residuals is used as a proxy for the variance of the noise in the Stokes and anti-Stokes intensity measurements. The consequence of a non-uniform temperature of the reference sections on the estimated variance is described in Section 10.

## 5. Single-Ended Calibration Procedure

In single-ended calibration, the temperature is estimated from Stokes and anti-Stokes intensity measurements with Equation (Equation 12). The parameters that need to be estimated from calibration are γ, Δα, and *C*, where *C* needs to be estimated for each time step. The parameters are estimated from the reference temperature at *M* locations along the reference sections and at *N* times. Equation (Equation 7) is reorganized to amend it for linear regression. The observation at location *m* and time *n*, denoted with $I_{m,n}$, is written as a linear combination of the unknown parameters:(26)$$\begin{array}{c} I_{m,n}=\frac{1}{T_{m,n}}\gamma -x_{m}\Delta \alpha -C_{n},\;\mathrm{with}\;m=1,2,\ldots ,M\;\mathrm{and}\;n=1,2,\ldots ,N \end{array}$$
where $T_{m,n}$ is the reference temperature at location *m* and time *n*, $x_{m}$ is the location of point *m* along the reference sections, and $C_{n}$ is the constant *C* of the fiber at time *n*. In total, there are N+2 unknown parameters and MN observations.

The system of *N* Equation (Equation 26) for location *m* may be written in vector form as:(27)$$\begin{array}{c} y_{m}=X_{m}a+\epsilon_{m}, \end{array}$$
where $\epsilon_{m}$ are the residuals between the observed values and the fitted values for location *m*, and
(28)$$\begin{array}{c} y_{m}=\left[\begin{array}{c} I_{m,1}\\ I_{m,2}\\ \vdots \\ I_{m,N} \end{array}\right],\;X_{m}=\left[\begin{array}{cccccc} \frac{1}{T_{m,1}} & -x_{m} & -1\\ \frac{1}{T_{m,2}} & -x_{m} & \; & -1\\ \vdots & \vdots & \; & \; & \ddots \\ \frac{1}{T_{m,N}} & -x_{m} & \; & \; & \; & -1 \end{array}\right],\;a=\left[\begin{array}{c} \gamma \\ \Delta \alpha \\ C_{1}\\ C_{2}\\ \vdots \\ C_{N} \end{array}\right] \end{array}$$

The vector a contains the unknown parameters that are to be estimated. The system of MN equations for all locations may be combined into one system of equations:(29)$$\begin{array}{c} y=Xa+\epsilon , \end{array}$$
where
(30)$$\begin{array}{c} y=\left[\begin{array}{c} y_{1}\\ y_{2}\\ \vdots \\ y_{M} \end{array}\right],\;X=\left[\begin{array}{c} X_{1}\\ X_{2}\\ \vdots \\ X_{M} \end{array}\right],\;\epsilon =\left[\begin{array}{c} \epsilon_{1}\\ \epsilon_{2}\\ \vdots \\ \epsilon_{M} \end{array}\right] \end{array}$$

This system (Equation (Equation 29)) is solved by minimizing the sum of the squared weighted residuals $\chi^{2}$: (31)$$\begin{array}{c} \chi^{2}={\left(y-Xa\right)}^{⊺}W\left(y-Xa\right) \end{array}$$
where ${}^{⊺}$ refers to the transposed matrix and W is a diagonal matrix given by
(32)$$\begin{array}{c} \mathrm{diag}\left(W\right)=\left[\begin{array}{c} W_{1}\\ W_{2}\\ \vdots \\ W_{M} \end{array}\right],\;W_{m}=\left[\begin{array}{c} \frac{1}{\sigma_{I_{m,1}}^{2}}\\ \frac{1}{\sigma_{I_{m,2}}^{2}}\\ \vdots \\ \frac{1}{\sigma_{I_{m,N}}^{2}} \end{array}\right] \end{array}$$

The variance, $\sigma_{I_{m,n}}^{2}$, of the distribution of the noise in the observation at location *m*, time *n*, is a function of the variance of the noise in the Stokes and anti-Stokes intensity measurements ($\sigma_{P_{+}}^{2}$ and $\sigma_{P_{-}}^{2}$), and is approximated with [24]: (33)$$\begin{array}{cc} \sigma_{I_{m,n}}^{2} & \approx {\left[\frac{\partial I_{m,n}}{\partial P_{m,n+}}\right]}^{2}\sigma_{P_{+}}^{2}+{\left[\frac{\partial I_{m,n}}{\partial P_{m,n-}}\right]}^{2}\sigma_{P_{-}}^{2} \end{array}$$
(34)$$\begin{array}{cc} \; & \approx \frac{1}{P_{m,n+}^{2}}\sigma_{P_{+}}^{2}+\frac{1}{P_{m,n-}^{2}}\sigma_{P_{-}}^{2} \end{array}$$

The variance of the noise in the Stokes and anti-Stokes intensity measurements is estimated directly from Stokes and anti-Stokes intensity measurements using the steps outlined in Section 4.

The sum of the squared weighted residuals may be minimized with a standard routine. In this paper, a custom sparse implementation of the weighted least squares routine of the Python package Statsmodels is used [25], which returns the optimal parameter set, the covariance matrix of the optimal parameter set, and the residuals (this is part of the Python package “dtscalibration”, which is described in Section 8). The number of reference sections required for calibration is discussed in Section 10.2.

## 6. Double-Ended Calibration Procedure

In double-ended calibration, the temperature is estimated using Stokes and anti-Stokes intensity measurements from both ends of the fiber. The temperature is estimated from Equations (Equation 16) and (Equation 17). The unknown parameters γ, $D_{F}\left(t\right)$, $D_{B}\left(t\right)$, and A(x) are obtained from calibration to reference sections, similar to the single-ended calibration procedure. $x_{1}$ is chosen as the first location of the first reference section, so that the value for *A* at that location equals 0 (Equation (Equation 15)). There are *M* locations along the reference sections where the temperature is measured at *N* times. In total there are 2N+M unknowns: parameter γ, *N* parameters $D_{F}$, *N* parameters $D_{B}$, and M−1 parameters *A* (since $A_{1}=0$). Equations (Equation 16) and (Equation 17) are reorganized to amend them for linear regression as:(35)$$\begin{array}{c} \left.\begin{array}{cc} I_{F,m,n} & =\frac{1}{T_{m,n}}\gamma -D_{F,n}-A_{m}\\ I_{B,m,n} & =\frac{1}{T_{m,n}}\gamma -D_{B,n}+A_{m} \end{array}\right\}\;\mathrm{with}\;m=1,2,\ldots ,M\;\mathrm{and}\;n=1,2,\ldots ,N \end{array}$$
where $D_{F,n}$ is the constant $D_{F}$ at time *n*, $D_{B,n}$ is the constant $D_{B}$ at time *n*$A_{m}$ is the constant *A* at location *m*, and $A_{1}=0$. The system of 2N equations for location *m* may be written as:(36)$$\begin{array}{c} y_{m}=X_{m}a+\epsilon_{m}, \end{array}$$
where
(37)$$\begin{array}{cc} \; & y_{m}=\left[\begin{array}{c} F_{m}\\ B_{m} \end{array}\right],\;X_{m}=\left[\begin{array}{cccc} R_{m} & -I(N,N) & 0(N,N) & -Q_{m}\\ R_{m} & 0(N,N) & -I(N,N) & Q_{m} \end{array}\right], \end{array}$$
(38)$$\begin{array}{c} F_{m}=\left[\begin{array}{c} I_{F,m,1}\\ I_{F,m,2}\\ \vdots \\ I_{F,m,N} \end{array}\right],\;B_{m}=\left[\begin{array}{c} I_{B,m,1}\\ I_{B,m,2}\\ \vdots \\ I_{B,m,N} \end{array}\right],\;R_{m}=\left[\begin{array}{c} \frac{1}{T_{m,1}}\\ \frac{1}{T_{m,2}}\\ \vdots \\ \frac{1}{T_{m,N}} \end{array}\right] \end{array}$$
where I is an identity matrix with its number of rows and columns between brackets, 0 is a matrix of zeros with its number of rows and columns between brackets, and $\epsilon_{m}$ are the residuals at location *m*. The matrix $Q_{m}$ is defined differently for the first location than for the other locations. For m=1, $Q_{m}$ is a matrix of zeros size *N* by M−1. For m>1, $Q_{m}$ is a matrix of zeros of size *N* by M−1 except for column m−1, which contains ones. The vector a contains the unknown parameters, has length 2N+M, and is given by:(39)$$\begin{array}{c} a=\left[\begin{array}{c} \gamma \\ a_{D,F}\\ a_{D,B}\\ a_{A} \end{array}\right],\;a_{D,F}=\left[\begin{array}{c} D_{F,1}\\ D_{F,2}\\ \vdots \\ D_{F,N} \end{array}\right],\;a_{D,B}=\left[\begin{array}{c} D_{B,1}\\ D_{B,2}\\ \vdots \\ D_{B,N} \end{array}\right],\;a_{A}=\left[\begin{array}{c} A_{2}\\ A_{3}\\ \vdots \\ A_{M} \end{array}\right] \end{array}$$

The weights of the observations at location *m* are given by:(40)$$\begin{array}{c} \mathrm{diag}\left(W_{m}\right)=\left[\begin{array}{c} 1/\sigma^{2}\left[F_{m}\right]\\ 1/\sigma^{2}\left[B_{m}\right] \end{array}\right] \end{array}$$
where the elements of $\sigma^{2}\left[F_{m}\right]$ and $\sigma^{2}\left[B_{m}\right]$ are approximated with Equation (Equation 34) applied to the forward and backward channels, respectively.

The equations for all *M* locations are gathered in a single set of equations given by Equations (Equation 29) and (Equation 30), where $y_{m}$, $X_{m}$, a are given by Equations (Equation 37)–(Equation 39). This is a system of 2MN equations for M+2N unknown parameters. The system is solved by minimizing $\chi^{2}$ as given by Equation (Equation 31), where W is given by Equation (Equation 32) with Equation (Equation 40) for $W_{m}$.

The set of estimated parameters contains estimates of *A* for the locations along the reference sections only. An estimate of *A* and its variance for locations outside the reference sections are required to estimate the temperature outside of the reference sections. An estimate of *A* for location *p* and time *n* outside the reference sections is obtained by setting $T_{F}=T_{B}$ using Equations (Equation 16) and (Equation 17), which gives:(41)$$\begin{array}{cc} A_{p,n} & =\frac{I_{B,p,n}-I_{F,p,n}}{2}+\frac{D_{B,n}-D_{F,n}}{2} \end{array}$$

The variance of $A_{p,n}$ is approximated with:(42)$$\begin{array}{cc} \sigma^{2}\left[A_{p,n}\right]\approx & \frac{1}{4}\left(\sigma^{2}\left[I_{B,p,n}\right]+\sigma^{2}\left[I_{F,p,n}\right]+\sigma^{2}\left[D_{F,n}\right]+\sigma^{2}\left[D_{B,n}\right]-2\sigma \left[D_{F,n},D_{B,n}\right]\right) \end{array}$$
(43)≈14(1PB,m,n+2σPB+2+1PB,m,n−2σPB−2+1PF,m,n+2σPF+2+1PF,m,n−2σPF−214(1PB,m,n+2σPB+2σPF−2+σ2DF,n+σ2DB,n−2σDF,n,DB,n)
where $\sigma_{P_{B}+}^{2}$, $\sigma_{P_{B}-}^{2}$, $\sigma_{P_{F}+}^{2}$, and $\sigma_{P_{F}-}^{2}$ are estimated directly from the Stokes and anti-Stokes intensity measurements of the forward and the backward channels using the procedure presented in Section 4, the terms $\sigma^{2}\left[D_{F,n}\right]$ and $\sigma^{2}\left[D_{B,n}\right]$ are parameter uncertainties from the diagonal of the covariance matrix, and the term $\sigma \left[D_{F,n},D_{B,n}\right]$ is the square-root of the covariance between $D_{F,n}$ and $D_{B,n}$ from the covariance matrix. A single estimate of $A_{p}$ and $\sigma^{2}\left[A_{p}\right]$ at location *p* is obtained using the inverse temporal-variance weighted mean: (44)$$\begin{array}{cc} A_{p} & =\sigma^{2}\left[A_{p}\right]\left(\sum_{n=1}^{N}\frac{A_{p,n}}{\sigma^{2}\left[A_{p,n}\right]}\right) \end{array}$$
(45)$$\begin{array}{cc} \sigma^{2}\left[A_{p}\right] & =\frac{1}{\sum_{n=1}^{N}1/\sigma^{2}\left[A_{p,n}\right]} \end{array}$$

The number of reference sections required for calibration is discussed in Section 10.2.

## 7. Confidence Intervals of the Temperature

The uncertainty in the estimated temperature varies along the fiber as the laser pulse attenuates when propagating through the fiber, and varies over time due to varying gains and losses in the DTS device. The two sources that contribute to the uncertainty in the temperature estimate are the uncertainty in the calibrated parameters and the uncertainty associated with the noise in the Stokes and anti-Stokes intensity measurements. The former dominates the uncertainty in the estimated temperature for measurements with longer acquisition times, while the latter dominates measurements with shorter acquisition times. Other sources of possible uncertainty are not taken into account here. These include the uncertainty introduced by the model that relates measured Stokes and anti-Stokes intensities to temperature, and the uncertainty in measured temperatures obtained with external sensors. The latter is generally much smaller than the uncertainty in the DTS temperature from the noise in the Stokes and anti-Stokes intensity measurements.

Estimation of the confidence intervals of the temperature starts with estimating separate probability density functions for the Stokes and anti-Stokes intensity measurements and the calibrated parameters. The probability density functions are propagated through the model using a Monte Carlo sampling procedure following the steps from JCGM [26], JCGM [27]. This procedure results in an approximation of the probability density function for the estimated temperature, which is different at each location and varies over time. Various summarizing statistics are computed from the approximate probability density function, including the expected value, the standard deviation, and the confidence intervals. The standard deviation is also called the temperature resolution, but in line with JCGM [26], the term standard uncertainty is used here. The procedure is explained first for single-ended measurements, followed by the procedure for double-ended measurements.

### 7.1. Single-Ended Measurements

Estimation of the confidence intervals for the temperatures measured with a single-ended setup consists of five steps. First, the variances of the Stokes and anti-Stokes intensity measurements are estimated following the steps in Section 4. A Normal distribution is assigned to each intensity measurement that is centered at the measurement and using the estimated variance. Second, a multi-variate Normal distribution is assigned to the estimated parameters using the covariance matrix from the calibration procedure presented in Section 5. Third, the distributions are sampled, and the temperature is computed with Equation (Equation 12). Fourth, step three is repeated, e.g., 10,000 times for each location and for each time. The resulting 10,000 realizations of the temperatures approximate the probability density functions of the estimated temperature at that location and time. Fifth, the standard uncertainties are computed with the standard deviations of the realizations of the temperatures, and the 95% confidence intervals are computed from the 2.5% and 97.5% percentiles of the realizations of the temperatures.

### 7.2. Double-Ended Measurements

Double-ended setups require four additional steps to estimate the confidence intervals for the temperature. First, the variances of the Stokes and anti-Stokes intensity measurements of the forward and backward channels are estimated following the steps in Section 4. A Normal distribution is assigned to each intensity measurement that is centered at the measurement and using the estimated variance. Second, a multi-variate Normal distribution is assigned to the estimated parameters using the covariance matrix from the calibration procedure presented in Section 6. Third, Normal distributions are assigned for *A* for each location outside of the reference sections. These distributions are centered around $A_{p}$ and have variance $\sigma^{2}\left[A_{p}\right]$ given by Equations (Equation 44) and (Equation 45). Fourth, the distributions are sampled and $T_{F,m,n}$ and $T_{B,m,n}$ are computed with Equations (Equation 16) and (Equation 17), respectively. Fifth, step four is repeated to compute, e.g., 10,000 realizations of $T_{F,m,n}$ and $T_{B,m,n}$ to approximate their probability density functions. Sixth, the standard uncertainties of $T_{F,m,n}$ and $T_{B,m,n}$ ($\sigma \left[T_{F,m,n}\right]$ and $\sigma \left[T_{B,m,n}\right]$) are estimated with the standard deviation of their realizations. Seventh, for each realization *i* the temperature $T_{m,n,i}$ is computed as the weighted average of $T_{F,m,n,i}$ and $T_{B,m,n,i}$:(46)$$\begin{array}{c} T_{m,n,i}=\sigma^{2}\left[T_{m,n}\right]\left(\frac{T_{F,m,n,i}}{\sigma^{2}\left[T_{F,m,n}\right]}+\frac{T_{B,m,n,i}}{\sigma^{2}\left[T_{B,m,n}\right]}\right) \end{array}$$
where
(47)$$\begin{array}{c} \sigma^{2}\left[T_{m,n}\right]=\frac{1}{1/\sigma^{2}\left[T_{F,m,n}\right]+1/\sigma^{2}\left[T_{B,m,n}\right]} \end{array}$$

The best estimate of the temperature $T_{m,n}$ is computed directly from the best estimates of $T_{F,m,n}$ and $T_{B,m,n}$ as:(48)$$\begin{array}{c} T_{m,n}=\sigma^{2}\left[T_{m,n}\right]\left(\frac{T_{F,m,n}}{\sigma^{2}\left[T_{F,m,n}\right]}+\frac{T_{B,m,n}}{\sigma^{2}\left[T_{B,m,n}\right]}\right) \end{array}$$

Alternatively, the best estimate of $T_{m,n}$ can be approximated with the mean of the $T_{m,n,i}$ values. Finally, the 95% confidence interval for $T_{m,n}$ are estimated with the 2.5% and 97.5% percentiles of $T_{m,n,i}$.

## 8. Python Implementation

The presented calibration procedure is implemented in the Python package “dtscalibration” [28]. It is open source, has a BSD 3-or-later license and is available online at https://github.com/dtscalibration/python-dts-calibration, together with installation instructions, examples, and documentation. The package reads DTS measurement files into an object, which has several calibration and plotting methods. Calibration and calculation of confidence intervals can be conducted within 10 lines of Python code, as is demonstrated in the examples of Section 9 and Appendix C. Several routines are implemented to read Stokes and anti-Stokes intensity measurement files from the following DTS devices: AP Sensing CP320, Sensornet Oryx, Sensornet Halo, SensorTran 5100, Silixa Ultima, Silixa XT. The package inherits many functions (e.g., visualization, parallel computing) from xarray [29] so that the code base remains small. Most computations are performed by Dask [30] in chunks and in parallel, so that gigabytes of DTS measurement data can be processed on a personal computer with limited memory.

## 9. Example

An example of a double-ended setup is presented to demonstrate the spatial and temporal variability in the uncertainty of the temperature. This example attempts to estimate the uncertainty of the temperature along the entire fiber for a given acquisition time. The uncertainty in the estimated temperature can easily be reduced by increasing the acquisition time, but that limits the ability to observe temporal variation in temperature. Alternatively, the uncertainty can be reduced by increasing the sampling distance, but that limits the ability to observe spatial variation in temperature.

### 9.1. Setup and Data Collection

A schematic representation of the setup is shown in Figure 2. Sections of the fiber are submerged in water baths with a measured temperature as listed in Table 1. A uniform temperature in all water baths is desired so that the temperature measured with an external temperature sensor resembles the temperature of the fiber. Therefore, an aluminum bath filled with water is placed in a cold climate room, and a second one is placed in a warm climate room. The air in the climate rooms circulates around the water baths and the air temperature is kept constant with a maximum variation of 0.5 °C to achieve a uniform temperature in the water baths. A cooler turns on if the air temperature in the climate rooms is 0.3 °C above its target temperature, and a heater turns on if the air temperature in the climate rooms is 0.2 °C below the target temperature. Both baths contain two coils of fiber. Another coil of fiber is placed in a cooler box filled with water, without temperature control, but with an aquarium air pump to attempt to mix the water and achieve a uniform temperature. All coils of fiber are lifted from the bottom of the baths with plastic spacers so that the coils are approximately centered. The setup was left to rest to achieve a stable and uniform temperature in the water baths before starting the measurements.

The DTS system used in this example is a Silixa Ultima-S DTS system (Hertfordshire, UK) that measures fibers up to 2 km with a factory-reported spatial resolution of approximately 30 cm. The system is configured to measure the Stokes and anti-Stokes intensity with an acquisition time of 2 s in the forward direction and 2 s in the backward direction, every 12.7 cm along 100 m of fiber (Figure 2). The system’s specification sheet lists several values for the standard uncertainty (referred to as the temperature resolution) for different single-ended setups using the built-in calibration routine. According to the specification sheet, the standard uncertainty is 0.34 °C if the fiber is shorter than 500 m, sampled every 12.7 cm, and measured for a second, which translates roughly to a standard uncertainty of 0.17 °C for an acquisition time of 4 s. The used fiber (j-BendAble made by j-fiber GmbH, Jena, Germany) is an OM3 fiber with a germanium-doped core with a diameter of 50 °m and a silica cladding with a diameter of 125 μm. The temperature in the warm- and cold-water baths is measured by the DTS system with two Pt100 RTDs, and the temperature in the ambient water bath is measured with a Pt100 RTD using a Fluke 1524 Handheld Thermometer.

### 9.2. Estimation of the Temperature and the Associated Uncertainty

The variance of the noise in the Stokes and anti-Stokes intensity measurements of the forward and backward channels were estimated using measurements from the two reference sections (‘Cold 1’ and ‘Warm 1’ in Table 1) with the procedure described in Section 4. A scatter plot with the residuals of the intensity measurements of the forward channel is shown in Appendix B. The residuals of the Stokes intensity measurements are plotted on the horizontal axis and the residuals of the anti-Stokes intensity measurements on the vertical axis. The residuals may be approximated with a Normal distribution and show no clear correlation (Pearson correlation coefficient is 0.02).

Double-ended calibration was performed to estimate the optimal parameter values and their covariance following the procedure described in Section 6. The calibration uses measurements from the two reference sections (Table 1). Instead of using all the measurements from the measurement period of one day, 1000 time samples were used, evenly spaced over a single day (N=1000). Inherent to DTS systems, neighboring measurement locations are spatially correlated (Section 2). To reduce this correlation, every other measurement location was disregarded so that the distance between the used measurement locations is 25.4 cm, which is close to the spatial resolution and results in a total of 193 measurement locations. The number of locations along the reference sections that are used for calibration is 76 (M=76).

The 95% confidence intervals are estimated with the approach described in Section 7. The estimated temperature and its confidence interval at the first time step are shown in Figure 3a. The difference between the estimated temperature and reference temperature is shown in Figure 3b with markers, where the confidence interval minus the estimated temperature at the first time step is shown with a blue fill. For the estimation of the confidence intervals, a Monte Carlo sample size of 10,000 was used; doubling the sample size for this measurement setup did not change the 95% confidence intervals significantly, but it did smooth the edges of the confidence intervals (i.e., the roughness of the edges of the blue fill in Figure 3b). The percentage of reference temperature measurements that fall within the 95% confidence interval of the estimated temperature for all 1000 time steps are listed for each section in Table 2. For the ambient bath, only 92.3% of the temperature measurements fall within the 95% confidence interval.

The mean difference between the reference temperature and the estimated temperature for all times is shown for each location along the reference sections in Figure 4 with orange markers. The estimated temperature of the ambient bath is on average 0.06 °C above the corresponding reference temperature. This bias may be the result of either the external temperature sensor not representing the temperature of the fiber due to local temperature variations in the water bath, or the external temperature sensor is calibrated differently. This could have been confirmed by simultaneously submerging all temperature sensors in a water bath and comparing their measurements, but this was not done here. The mean difference of reference section Warm 2 shows a spatial correlation between the mean differences, which can indicate non-uniform temperatures in the water bath. The spatial variability in the standard deviation of the differences between the reference temperature and the estimated temperature is shown in Figure 4 with blue markers. The standard uncertainty is estimated for each location from the Monte Carlo samples from all times and is shown with a black line. It varies with the temperature, with slightly smaller values for the fiber sections in the cold baths and slightly larger values for the fiber sections in the warm baths. The standard deviation of the difference between the reference temperature and the estimated temperature for all times is shown with blue markers and is well described at each location with the standard uncertainty.

The mean difference between the reference temperature and the estimated temperature for all reference locations is plotted over time in Figure 5 with orange markers. Each marker represents the mean of 193 differences. The mean of the differences are slightly above zero probably caused by the bias in the ambient bath temperature. The standard uncertainty at each time is computed from the Monte Carlo samples from all locations and is shown with a black line. The standard deviation of the difference between the reference and estimated temperature for all locations is shown with blue markers, which roughly follow the estimated standard uncertainties. The increase in standard uncertainty after 11.5 h coincides with a strong decrease in Stokes and anti-Stokes intensity of the forward and backward channels (not shown here), caused by either a decrease in laser strength or a decrease in sensitivity of the detector.

### 9.3. Effect of Parameter Uncertainty

As stated, the uncertainty of the temperature is a function of the uncertainty in the estimated parameters and the uncertainty in the Stokes and anti-Stokes intensity measurements. The estimation of the standard uncertainty of the temperature was repeated except for that the covariance matrix of the optimal parameters was not propagated to the Monte Carlo set, so that the standard uncertainty consists of only the uncertainty introduced by the noise in the Stokes and anti-Stokes intensity measurements (Section 7.2). The standard uncertainty of the temperature was recalculated, and was on average 0.001 °C smaller than the original values and did not show any patterns or trends. Hence, it is concluded that for this experiment the contribution of the parameter uncertainty to the standard uncertainty is small compared to the uncertainty introduced by the noise in the Stokes and anti-Stokes intensity measurements.

### 9.4. Effect of Difference in Reference Temperatures

As stated in Section 6, reference sections at two different temperatures are required to estimate the parameters that relate Stokes and anti-Stokes intensity measurements to temperature and only one reference section is needed if its temperature varies sufficiently over time. To test the consequence of calibrating to reference sections with a variation in temperature that is too small, an additional calibration is performed with two reference sections at the same temperature: Cold 1 and Cold 2 (Table 1). 95% of the reference temperatures were between 4.33 and 4.39 °C. Values for the A(x) parameters along the reference sections are correctly estimated, but the parameters γ, $D_{F}\left(t\right)$, and $D_{B}\left(t\right)$ are not. Their estimated values are of the right order of magnitude, and the temperature of the cold-water baths is estimated well with those parameters (not shown). The temperatures outside the 4.33–4.39 °C range are not correctly estimated. The 95% confidence intervals for the estimated temperature outside the reference sections are enormously wide, because of the large parameter uncertainty. The standard uncertainty of the A(x) estimates are the same as in the first calibration configuration, but the standard uncertainty of the γ, $D_{F}\left(t\right)$, and $D_{B}\left(t\right)$ estimates are two orders of magnitude larger than their estimated values. As expected, this calibration configuration failed due to the lack of difference in temperature between the reference sections and the lack of temperature variation of the reference sections.

## 10. Discussion

### 10.1. Improved Temperature Estimation for Double-Ended Setups

The estimation of the uncertainty of the temperature from Stokes and anti-Stokes intensity measurements requires several approximations. In this section, the practical implications of these approximations are assessed. Single-ended measurements have several inherent drawbacks compared to double-ended measurements. The differential attenuation is approximated to be constant, step changes are neglected (e.g., fiber splices, sharp bends), and fiber sections with a different differential attenuation are not accounted for (e.g., coiled up fiber sections, different fiber types). This includes sharp bends in ‘bend-tolerant’ fiber [14,31]. The step losses in Stokes and anti-Stokes intensity measurements can be corrected manually, if they are identified and the temperature is the same at either side of the step loss [32]. Uncorrected step losses result in a bias in the estimated temperature that is not expressed in the confidence intervals. Commercially available DTS systems usually include an internal reference fiber section for calibration purposes. The use of the internal reference section is not recommended for single-ended measurements because of the step change in the integrated differential attenuation at the connector. The type of the internal fiber is most likely different, with a different differential attenuation than the fiber connected to the DTS system. Hence, it is recommended to use double-ended setups rather than single-ended setups.

The proposed calibration procedure to estimate the integrated differential attenuation in double-ended setups differs from the two-step procedure of van de Giesen et al. [12]. They estimate the differential attenuation integrated between neighboring measurement locations, average them over time for the entire fiber, and sum them to estimate the integrated differential attenuation. In Section 6 of this paper, estimation of the integrated differential attenuation is achieved in one step, and formulated so that it can be used in weighted linear regression.

The presented calibration procedure can lead to better temperature estimates compared to existing calibration procedures. Parameters that are time invariant are kept constant and are estimated from all available data, which improves their estimation. By weighing Stokes and anti-Stokes measurements with their uncertainty, the estimation of parameters is less affected by measurement noise along reference sections with a low signal strength, i.e., at the end of the fiber.

Furthermore, weighted averaging of the temperature from the forward and backward-channel measurements results in better temperature estimates than arithmetic averaging or using either of the individual channel measurements. Consider the synthetic temperature measurement of a double-ended setup shown in Figure 6. The approximated standard uncertainty is computed using the procedure outlined in Section 7.2 and is shown for the forward-channel temperature measurements in orange and for the backward channel in blue. The standard uncertainty of the inverse-variance weighted mean is shown with the solid black line and is much smaller along the entire fiber.

### 10.2. Calibration to Reference Sections

The presented calibration procedures require either one or two reference sections with a different temperature. In single-ended setups, two reference sections are needed to differentiate between $\gamma /T_{r}$ and C(t)+xΔα. In double-ended setups, two reference sections are needed to differentiate between $\gamma /T_{r}$ and $D_{F}\left(t\right)$, and between $\gamma /T_{r}$ and $D_{B}\left(t\right)$. Only one reference section is needed in single-ended and double-ended setups if the reference temperature varies sufficiently over time. To examine the minimum temperature variation that is needed when using one reference section, the analysis of the example of Section 9 is repeated using only the reference section in the ambient water bath, which varies between 12.53 and 12.73 °C. This temperature variation of only 0.2 °C proved to be sufficient for this setup. The contribution of the parameter uncertainty to the standard uncertainty is again small com1pared to the uncertainty introduced by the noise in the Stokes and anti-Stokes intensity measurements. As expected, the likely bias in the estimated temperature of the ambient bath that was apparent in Figure 4 translates to a bias in the estimated temperature (not shown). For example, the temperature estimates along the sections in the cold and warm water baths are 0.03 and 0.10 °C too low, respectively. The minimum temperature difference needed for calibration depends on many factors, including: the matrix solver and its settings, the length of the reference sections, the Stokes and anti-Stokes intensity, and the noise in the Stokes and anti-Stokes intensity measurements.

All existing calibration procedures rely on the temperature of reference sections measured with external sensors. Any deviation of the fiber temperature from the temperature measured with an external sensor introduces errors. Problems that are commonly occurring with DTS calibration are ill-defined positions of the reference sections and non-uniform temperature of the water baths. Both can be discovered at different stages of the calibration procedure. They may introduce a bias in the estimated temperature, which is not accounted for in the estimation of the standard uncertainty and confidence intervals. Therefore, an additional external temperature sensor is recommended for identification of a bias in the reference temperature. A bias may then be identified by, for example, composing a figure of the time-averaged difference between the estimated temperature and the reference temperature, similar to Figure 4. A non-uniform temperature of the reference sections also negatively affects the parameter estimation by introducing an error in the coefficient matrix, in the first column of $X_{m}$ in Equations (Equation 30) and (Equation 37). Investigation of the contribution of the parameter uncertainty to the uncertainty of the estimated temperature is explained in Section 9.3. A non-uniform temperature of the reference sections may also result in an overestimation of the noise in the Stokes and anti-Stokes intensity measurements. The variance of the noise in the intensity measurements are estimated from the residuals between Equations (Equation 20) and (Equation 21) and the intensity measurements assuming a uniform reference temperature (Section 4). A non-uniform temperature of a reference section increases the residuals, which are incorrectly attributed to the noise from the detector. A comparison of the estimated variance of different reference sections can indicate the location of non-uniform temperature sections. Alternatively, trends in the time-averaged differences between the estimated and reference temperature can also indicate a non-uniform temperature of the reference sections, of which an example is shown with orange markers in Figure 4.

## 11. Conclusions

A new approach is presented to calibrate temperature from Stokes and anti-Stokes intensity measurements and to provide a confidence interval for the estimated temperature. The uncertainty in the estimated temperature is caused by the noise from the Stokes and anti-Stokes detectors, and the uncertainty in the calibrated parameters that relate the Stokes and anti-Stokes intensity measurements to temperature. Estimation of the confidence interval for the estimated temperature requires an estimation of the distribution of the noise from the Stokes and anti-Stokes detectors and a multi-variate distribution of the parameters that relate the Stokes and anti-Stokes intensity measurements to temperature. All these distributions are propagated with Monte Carlo sampling to approximate the probability density function of the temperature, which is different at each location and varies over time. Various summarizing statistics can be computed from the approximated probability density function, such as standard uncertainties and confidence intervals.

Several improvements were made to existing calibration procedures to reduce the uncertainty in the estimated temperature. The integrated differential attenuation differs per setup and must be experimentally estimated to differentiate it from a shift in temperature. The integrated differential attenuation for double-ended setups is formulated such that it can be used directly in linear regression and the uncertainty in the parameter estimation is reduced. The parameter estimation is further improved by making use of the fact that some parameters are time-invariant, and to use all available data to estimate their value. The linear regression accounts for spatial and temporal variation in the signal intensity, so that large-intensity measurements have a larger weight in the parameter estimation than the small-intensity measurements.

In double-ended setups, the temperature is estimated from measurements made from both ends of the fiber. Close to the ends of the fiber, the difference in Stokes and anti-Stokes intensity between the forward and backward measurements is large, resulting in a large difference in the uncertainty of the temperature estimated from the forward and backward measurements. The estimated temperature is a weighted average of the temperatures estimated from the forward and backward-channel measurements. Compared to unweighted averaging, this reduces the uncertainty in the estimated temperature. The uncertainty of the temperature that is estimated with the proposed calibration procedure is assessed in an example. The estimated temperature and 95% confidence intervals adequately represent the temperature of the reference sections.

The calibration procedure is implemented in “dtscalibration”, an open-source Python package that is freely available under the BSD 3-or-later license from https://github.com/dtscalibration/python-dts-calibration. The package contains the new calibration procedures for both single-ended and double-ended setups, can compute confidence intervals of the estimated temperature, and includes several routines for visualization of the results.
Dataset: https://doi.org/10.4121/uuid:71b5c3c2-4105-4f4f-bd1e-d7c56732a665Dataset license: GPL-3.0-or-later

## Figures and Tables

**Figure 1 sensors-20-02235-f001:**
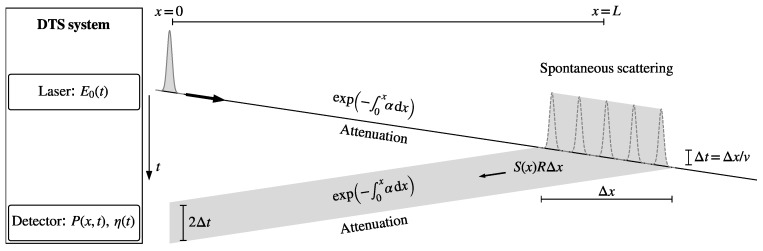
Path of a laser pulse.

**Figure 2 sensors-20-02235-f002:**
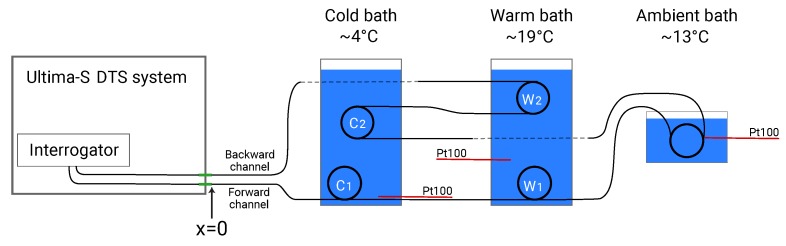
The setup of the example.

**Figure 3 sensors-20-02235-f003:**
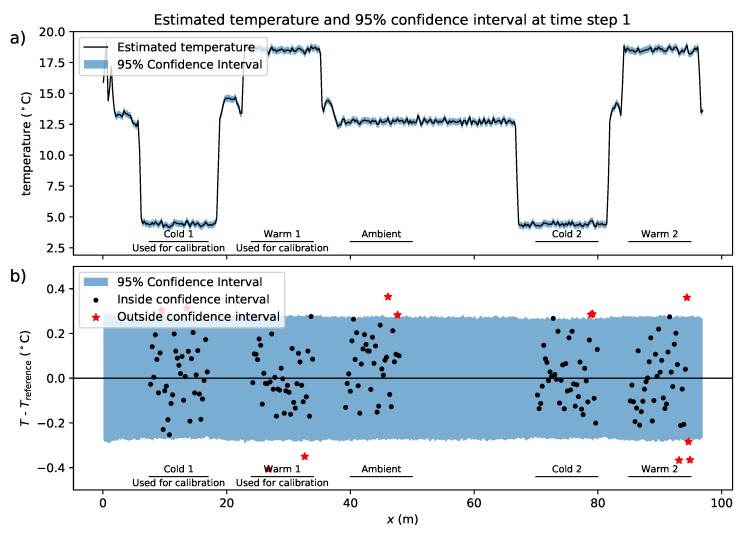
(**a**) Temperature with its 95% confidence intervals at the first time step. (**b**) Differences between the estimated temperature and the reference temperature at the first time step.

**Figure 4 sensors-20-02235-f004:**
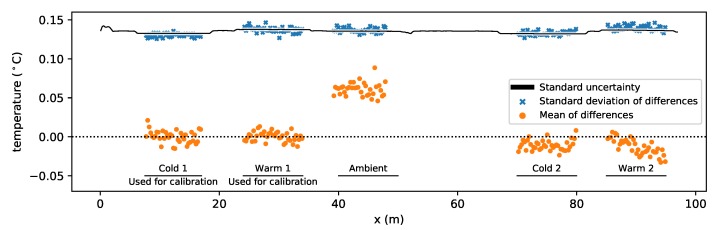
Spatial variation of the standard uncertainty of the estimated temperature, and the mean and standard deviation of the differences between the estimated and reference temperature.

**Figure 5 sensors-20-02235-f005:**
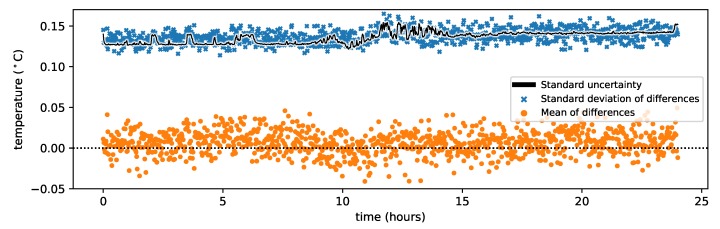
Temporal variation of the standard uncertainty of the estimated temperature, and the mean and standard deviation of the differences between the estimated and reference temperature.

**Figure 6 sensors-20-02235-f006:**
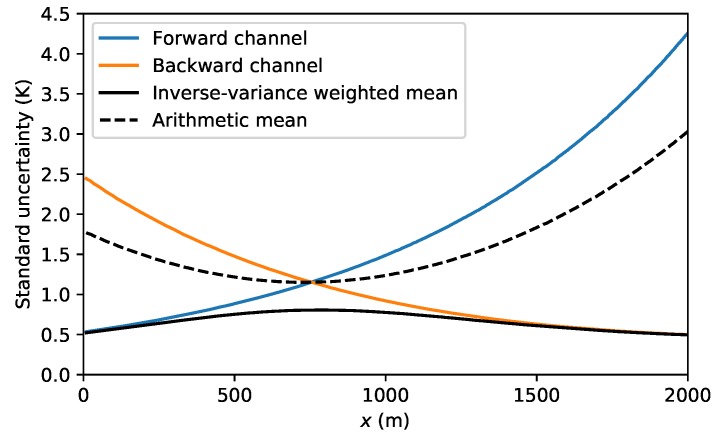
Synthetic example of the standard uncertainty of the estimated temperature using arithmetic mean and the inverse-variance weighted mean.

**Table 1 sensors-20-02235-t001:** Fiber sections submerged in water baths.

Name	Fiber Section (m)	Average Temperature (°C)	Number of Measurement Locations	Notes
Cold 1	7.5–17.0	4.35	37	Used for calibration
Warm 1	24.0–34.0	18.52	39	Used for calibration
Ambient	40.0–50.0	12.62	39	
Cold 2	70.0–80.0	4.35	39	
Warm 2	85.0–95.0	18.52	39	

**Table 2 sensors-20-02235-t002:** Percentage of reference temperature within estimated 95% confidence intervals.

Cold 1	Warm 1	Ambient	Cold 2	Warm 2	Total
95.6%	95.0%	92.3%	94.7%	94.3%	94.4%

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
