# Peer review of "Estimation of Temperature and Associated Uncertainty from Fiber-Optic Raman-Spectrum Distributed Temperature Sensing"

_sensors, 2020, doi:10.3390/s20082235_

Round 1

Reviewer 1 Report

In their article, the authors presented a novel procedure for the temperature estimates for single-ended and double-ended setups. The article also includes practical aspects of how to reduce the uncertainty in the temperature. Their procedure includes an estimation of the uncertainty in the form of confidence intervals. The authors provide full forward and backward channel measurement analysis. I find the subject of the paper relevant to the field of “MDPI sensors” of importance for the scientific community and suitable for publication in “MDPI sensors”.

The overall evaluation of the study is positive. Moreover, taking into account the originality and the usefulness of the conducted research, it is worth (though it is not obligatory) to consider including in the article some additional information in line with the following questions and suggestions:

  1. Section 2. Authors note that the constant g in equations (2) and (3) is the sensitivity of the Stokes scattering to temperature in Kelvin, but they did not give its sample value.
  2. Line 96: The same remark applies to the attenuation coefficients a+ and a-.
  3. What is the name of C? This should be completed.
  4. The equations are part of the sentence, so there seems to be a lack of punctuation after each equation. For example (“.” or “,” or “;”).
  5. Authors should also complement colons before some equations, e.g.: (2), (3), (6), (7), (8), (10), (11), (12), (16), (18), (26), (27)-(32), (35), (36), (39)-(48).
  6. The formula described by (12) should be written using “≈” instead of “=”.
  7. The vector/matrix a used in equation (28) is not explained (i.e. not described). Please complete it behind this formula.
  8. Just a small note: using n to mark time instead of t (section 4) is a bit confusing, but please don't change it anymore. In articles on this subject, there are also markings other than t, most often n.
  9. Where does the number of iterations equal to 10,000 results from? That amount was enough in single-ended and double-ended measurements?

The additional materials in the form of source codes allow to verify the operation of the algorithm and are undoubtedly the additional positive point of this publication. The paper contains many interesting metrological aspects related to distributed temperature measurements.

This work contains a lot of research material and I definitely think that its publication in MDPI Sensors corresponds to the subject of this journal.

My conclusions:

The subject of the paper, presented results and conclusions are of importance for the scientific community interested in DTS measurement systems. I have marked minor corrections, but they do not affect my positive assessment of the manuscript.

Reviewer 2 Report

In this paper, a new calibration approach is proposed for Distributed temperature sensing (DTS) system. Various summarizing statistics, about the estimated temperature, which varies along the fiber and with time, are computed from the approximate probability density function, such as the confidence intervals and the standard uncertainty. Experimental validation confirms the proposed approach.

The subject is very well introduced, including a good bibliography. The DTS the principle and the uncertainties about the temperature measurement are well detailed. I find that the paper is very clear in its structure and well written. The paper is complete. Maybe sometimes there is a lack of synthesis. I also think that the authors could insist more on the contribution of their solution compared to the bibliography. Does the proposed calibration method make a significant jump in the reliability of temperature measurement?

This paper can be accepted.

- Figure A1 must be rename B1

Reviewer 3 Report

Review of draft ‚Estimation of temperature and associated uncertainty from fiber-optic Raman-spectrum distributed temperature sensing‘ by des Tombe et al., submitted to Sensors, 738700

General comment:

This is a refreshingly novel manuscript which can help moving the DTS community in the direction it should with regard to calibrating FO-DTS measurements: away from the basic optics-based calibration equation solely accounting for the classic three calibration parameters $\gamma, C(t) and \Delta \alpha x$, to one which explicitly  factors in the instrument noise. Current calibration routines adjust the three calibration parameters at each time step resulting in significant temporal variability in these statistically interdependent parameters where there should be none, e.g. $\gamma$. The need to have a new calibration approach specifically for less spatially and temporally integrated observations aiming at resolving the thermal structure of fast evolving fluid flows has been growing over the last years specifically with the acceptance of fiber-optic distributed sensing (fods) techniques into the atmospheric sciences. The scientific goal to separate the true turbulence signal from the instrument noise becomes critical in order to resolve the finest flow vortices. Hence, this new calibration approach is much appreciated and needed.

The manuscript is well written with a clear technically and mathematically inclined audience in mind, it reads well, and is concise. I do have a few mostly (minor) comments which can help shaping the manuscript for the reader. Of course, ideally one would include a more evolved section to demonstrate the utility of this new calibration approach to separate the geophysical signal of interest from the instrument noise, but the scope of current section 9 is fine given the technical nature of this manuscript. I recommend its acceptance after administering minor revisions.

Minor comments:

  1. Lines 117ff: This can only be accomplished if each of the individual fiber sections have the required reference sections. This needs to be stated here to avoid false impression for the novice user. Unless the authors intend to make use of duplexed sections in case those exist.
  2. Actually, it would be beneficial to include information about the utility of duplexed cable configurations for both the single- and doubled-ended setups. I imagine those could serve as some sort of ambient validation bath similar to the one described in Section 9, but with significant uncertainty of the absolute temperature. This physical fiber configuration of some duplexed section with differences in x and hence differential attenuation, but with the same temperature within the fiber-optic array is easy to accomplish in any deployment. Such a section often happens to be in the array anyway because of routing the fiber-optic cable from/ to the DTS instrument to the measurement location. Why not make use of it?
  3. Line 123: I advise against the use of the letter ‘B’ to refer to the backward channel in double-ended configurations to avoid confusions. B is already used in Eqs. 1,4,5. I recommend using subscript ‘R’ for reverse.
  4. Lines 132ff: While this is a necessary assumption, it’s almost always satisfied given the small temporal separation of subsequent measurements with short integration times and slower-response cables on the order of seconds. Maybe this should be reformulated as a recommendation in such a fashion that the user needs to design the sampling sequence such that the forward and reverse signals are separated by only a fraction of the time constant of the fiber-optic cable.
  5. 17: Based upon Eqs. 14, 15 and 19, I believe the minus sign in the last term of the denominator needs to read ‘plus’, since introducing the term D_B(t) in Eq. 19 only splits the total integral of the differential attenuation between L and x into two sections: L to x1, and x1 to x. If I am wrong in my assumption, then please explain why A(x) needs to be subtracted. In addition, it would be helpful to mention at this point how you chose x_1 (stated only later), so the reader understands why to split the differential attenuation into the two mentioned sections.
  6. Line 161f: This statement seems counter-intuitive and prompts an explanation.
  7. Line 169f: You describe in section 10 how non-uniform temperature distributions in the reference sections, which are very common in most water bath designs, affect the calibration uncertainty. Maybe include a note about/ reference to this here?
  8. 20-25: I do appreciate the attempt of splitting the signal into time-variant and space-invariant, and time-invariant and space-variant components G and H. However, I recommend dropping this separation and corresponding equations for reasons of a) being highly correlated with each other as you correctly state, and are evaluated as a convolute, and b) H in practice also varies with time because of thermal expansion of fibers with temperature changes and some bends etc. Hence, the separation seems artificial and does not add any insights into the calibration procedure.
  9. Line 193: Mention which residuals you referring to here.
  10. Example in Line 201, but throughout: General remark: I do not like how both Raman-signals are referred to collectively by ‘(anti-) Stokes’. Parentheses should be avoided as they suggest difference in importance. Is there a better way w/o parentheses? How about simply ‘Anti-/ Stokes’?
  11. Lines after 237 before Eq. 41: This is a confusing statement which can easily be misunderstood since this is the actual goal of the entire calibration process, correct? I think you meant to say you need one more section of known temperature (e.g. your ambient bath in section 9). Please clearly separate the measurement array from the needed calibration sections.
  12. Lines 246ff: Please specify the term' varies sufficiently'. Is certainly has implications for estimating the time scale of the noise.
  13. Line 320ff: It is unclear to me what the 0.5 \circ C is referring to: accuracy of the bath temperature, precision of the bath temperature, or allowed spatial variability of the bath temperature? This has important implications for the actual calibration routine.
  14. Figure 2: Please remove the brand/model from the figure. DTS machine would be more general.
  15. Line 327: Please state that you are using the 2km variant of the Ultima-S machine to distinguish it from the 5 km variant, which has a much-increased noise floor. It’s important to not confuse the reader.
  16. Line 337ff: Wouldn't it be nice if one included error in reference bath temperature as well? In actual installations this is often an issue particularly when the PT-100s supplied with the DTS machine are read with the DTS-internal electronics. Those are subject to noise as well. I believe that this is one additional component in the modeled noise in this new calibration routine, correct? If yes, then please include it from the beginning.
  17. Line 371ff: Well, again, this source of uncertainty needs to be included in the discussion, please see my earlier comments. Was the handheld fluke immersed in the water bath read by the DTS Pt-100s? This could have been easily checked.
  18. Line 388, Figure 5: This sudden change in variance completely confusing and has nothing to do with demonstrating the utility of the calibration routine, which is at the heart of this manuscript. It should be dropped from the displayed data. If it is a common feature of this specific setup or DTS machine, then it needs to be discussed elsewhere in a manuscript which is dedicated to summarizing those artifacts.
  19. Line 416ff: could this discussion be expanded to a systematic analysis, ie, which minimum \Delta T between reference bath temperatures is needed to correctly calibrate temperatures outside of the reference baths and in the measurement section? This current recommendation to users is: reference bath temperatures should span the range of expected ambient temperatures. But I wonder if this new calibration routine would give some additional insights. In certain extreme environments it is difficult to maintain reference bath temperatures because of immense heating/ cooling requirements.

Reviewer 4 Report

DTS has been widely used for many applications. The system measures the backscatter intensity which, however, causes uncertainties of temperature measurement. The cable, the environment, the connections between fiber and system and others may cause significant errors on the measurement. The authors provide a new calibration method to improve the resolution of temperature measurement, illustrate the method in details and give an example on the calibration result. The manuscript is certainly within the scope of this journal and has no obvious defect. I would recommend the publication and look forward to the potential use of the Python package. Finally, a small question is that, the authors used a Silixa Ultima-S DTS system, which is an advanced equipment with high temperature resolution and stable measurement, in the example. Will the performance of the device affect the calibration result since calibration is more important for such devices?
